# Acute Impact of Polyphenol-Rich vs. Carbohydrate-Rich Foods and Beverages on Exercise-Induced ROS and FRAP in Healthy Sedentary Female Adults—A Randomized Controlled Trial

**DOI:** 10.3390/antiox14121481

**Published:** 2025-12-10

**Authors:** Markus Gassner, Laura Bragagna, Helia Heidari Dasht Bayaz, Caroline Stumpf-Knaus, Laura Schlosser, Julia Lemberg, Julia Brem, Marc Pignitter, Matthias Strauss, Karl-Heinz Wagner, Daniel König

**Affiliations:** 1Department of Nutrition, Faculty of Life Sciences, Section for Nutrition, Exercise and Health, University of Vienna, 1090 Vienna, Austria; 2Vienna Doctoral School of Pharmaceutical, Nutritional and Sport Sciences, University of Vienna, 1090 Vienna, Austria; laura.bragagna@univie.ac.at; 3Department of Nutritional Sciences, University of Vienna, 1090 Vienna, Austriakarl-heinz.wagner@univie.ac.at (K.-H.W.); 4Institute of Physiological Chemistry, Faculty of Chemistry, University of Vienna, Josef-Holaubek-Platz 2, 1090 Vienna, Austriamarc.pignitter@univie.ac.at (M.P.); matthias_strauss@univie.ac.at (M.S.); 5Vienna Doctoral School in Chemistry (DoSChem), Faculty of Chemistry, University of Vienna, Währinger Str. 42, 1090 Vienna, Austria; 6Centre for Sport Science and University Sports, Department of Sports Science, Section for Nutrition, Exercise and Health, University of Vienna, Auf der Schmelz 6, 1150 Vienna, Austria

**Keywords:** ROS, exercise-induced oxidative stress, polyphenols, resistance exercise, HIIT, pomegranate juice, blueberries, FRAP, carbohydrates

## Abstract

Polyphenols and carbohydrates may modulate exercise-induced oxidative stress through distinct mechanisms: polyphenols via antioxidative properties, and carbohydrates via improved rapidly available energy supply. This randomized controlled trial (RCT) compared the acute effects of polyphenol-rich foods (pomegranate juice, blueberries), carbohydrate-rich foods (whole-grain bread, bread roll) and water control on HIIT-induced changes in ROS and FRAP in thirty healthy females. We conducted an RCT with two parallel intervention cohorts (study arm one: blueberries, whole-grain bread, bread roll, water (control); study arm two: pomegranate juice, water (control)), in which participants completed single-dose intervention days following 12 h fasting. On each intervention day, ROS and FRAP were assessed at baseline, pre-HIIT, post-HIIT and after 15 min recovery. Carbohydrate intake significantly reduced FRAP elevations (−2.16% (*p* < 0.05)) vs. polyphenols (−2.49% vs. water (*p* < 0.01) from pre-HIIT to post-HIIT). Furthermore, carbohydrate intake showed a tendency toward attenuating the exercise-induced increase in ROS (−7.75%, *p* = 0.095, vs. polyphenols from pre-HIIT to post-HIIT), although this did not reach statistical significance. Polyphenol-rich foods accelerated ROS reduction during the 15 min recovery phase (−8.22% (*p* < 0.01) vs. carbohydrates). No significant differences compared to water were observed from pre-HIIT to post-HIIT (polyphenols: *p* > 0.05; carbohydrates: *p* > 0.05) or from post-HIIT to 15 min post-HIIT (polyphenols: *p* > 0.05; carbohydrates: *p* > 0.05), which could be expected under fasted HIIT conditions. Overall, carbohydrates mitigated oxidative stress during exercise, whereas polyphenol-rich foods supported short-term post-exercise recovery.

## 1. Introduction

Exercise-induced oxidative stress is well established in sports nutrition research, yet the exact roles of reactive oxygen species and antioxidative biomarkers remain controversial. ROS can exert both harmful and beneficial effects, depending on their concentration and timing [1,2]. Excessive ROS production has been associated with impaired muscle recovery, contractile dysfunction, muscle soreness and, in the long term, compromised immune function [3,4]. Chronically elevated oxidative stress markers are also implicated in the pathogenesis of various non-communicable diseases, including cardiovascular disease, type 2 diabetes, DNA damage and aging-related conditions [5].

Conversely, physiological levels of ROS may play essential roles in initiating beneficial adaptations to exercise, such as muscle hypertrophy and improved antioxidant endogenous defense mechanisms [6,7]. This dual role suggests the existence of an ideal physiological state, where moderate levels of oxidative stress and inflammation optimize both exercise performance and recovery.

In this context, two primary nutrient groups have been identified as having the potential to attenuate exercise-induced oxidative stress: carbohydrates and polyphenols.

First, carbohydrate intake prior to exercise has been shown to mitigate oxidative stress responses. Specifically, sucrose—a disaccharide abundantly found in plant leaves—has demonstrated the ability to scavenge hydroxyl radicals in vitro [8].

Similarly, diets enriched in glucose have been associated with increased resistance to oxidative damage [9], and both glucose and sucrose have been reported to efficiently neutralize hydroxyl radicals [10].

Furthermore, carbohydrate consumption has been reported to reduce the full-body stress response to exercise. A review by Nieman points out that carbohydrate ingestion during extended endurance exercise, such as marathon running or triathlon, is linked to attenuated physiological stress responses, including reduced cortisol and growth hormone release, diminished disruption in immune cell counts and lower pro-inflammatory cytokine response compared to placebo [11,12].

Further supporting evidence for a favorable effect of carbohydrate consumption was shown by studies, which have demonstrated that mitochondria oxidizing fatty acids produce significantly more ROS than when using glucose-derived substrates such as pyruvate. This suggests that favoring carbohydrate metabolism over fatty acid oxidation may help reduce ROS formation during oxidative phosphorylation [13].

We aimed to test two high-carbohydrate (whole-grain bread, bread roll) sources with different glycemic indices (GIs) and fiber contents (Table 1) in order to detect potential differences in absorption and/or glycemic response.

Polyphenols, a diverse class of phytochemicals, have also received considerable attention for their antioxidant and anti-inflammatory properties [14,15]. Their characteristic aromatic structures and conjugated double bonds confer strong free radical scavenging capacity [16,17,18].

In vitro studies have shown that polyphenols are stronger antioxidants than conventional antioxidants such as vitamins C and E [19].

Early approaches to mitigating oxidative stress often focused on high-dose vitamin supplementation. However, these interventions were found to suppress endogenous antioxidant enzyme responses, highlighting the limitations of isolated high-dosage nutrient interventions [20,21].

In contrast, whole foods rich in antioxidants and anti-inflammatory compounds are increasingly recognized as a more physiologically relevant and effective approach [22,23,24].

Furthermore, we selected resistance-circuit HIIT as the stimulus to generate a physiological stress response because it remains less explored than conventional aerobic HIIT or traditional resistance training. To our knowledge, the only resistance-based circuit protocols that have elicited a measurable stress response are those from Zeng and Bizheh [25,26]. These protocols inspired the design of the resistance-circuit HIIT model applied in the presented study.

Given the limited evidence from acute, food-based interventions, this RCT investigated whether polyphenol-rich or carbohydrate-rich foods and beverages differently influence resistance-circuit high-intensity interval training (HIIT)-induced alterations in ROS and FRAP in healthy sedentary young female adults. We combined the data for certain foods (high-polyphenol: pomegranate juice/blueberries; high-carbohydrate and low-polyphenol: bread roll/whole-grain bread) to potentially provide novel insights into whether differences in fiber content or polyphenol content may influence recovery after exercise.

## 2. Materials and Methods

### 2.1. Study Design

This study was conducted as a randomized, controlled, crossover acute intervention trial with two parallel intervention cohorts with a washout period of at least 7 days between intervention days. The primary objective was to investigate the acute effects of multiple interventions on ROS and FRAP responses.

In total, there were five possible treatments: 1× pomegranate juice, 1× blueberries (category: polyphenol-rich), 1× whole-grain bread, 1× bread roll (category: low-polyphenol, high-carbohydrate) and 1× water control. The aim was to examine the effects of these treatments on ROS and FRAP responses induced by resistance-circuit high-intensity interval training (HIIT) in healthy sedentary female participants. Water intake served as the control condition.

On each intervention day, venous and capillary blood samples were collected at four standardized time points: (1) baseline (after 12 h overnight fasting), (2) pre-HIIT (120 min after food consumption, immediately prior to HIIT); (3) post-HIIT (immediately following the HIIT session) and (4) 15-post-HIIT (after 15 min of recovery).

The primary outcome measures included ROS (capillary blood, electron paramagnetic resonance spectroscopy) and ferric reducing ability of plasma (FRAP, plasma).

Throughout the study period, participants were instructed to refrain from using dietary supplements and to maintain their habitual dietary and physical activity routines, with physical activity not to exceed 2 h per week.

The participants had to perform a 24 h food recall before each intervention day.

This study received ethical approval from the Ethics Committee of the University of Vienna (Approval ID: 00743-1.12.2021) and was registered on ClinicalTrials.gov under the identifier NCT05242978 (Registration date: 26 January 2022).

The detailed study design and blood sampling schedule are illustrated in the graphical abstract.

### 2.2. Study Participants

Out of 83 individuals screened for eligibility, 45 sedentary female participants, aged 19–33 years, were randomly assigned to study arm one or two, with a random order of consumption of each of the included interventions to minimize the possibility of a time effect confounding the results. We wanted to avoid the possibility that every participant consumed her interventions in the same order (e.g., study arm one: always (1) bread roll, (2) whole-grain bread, (3) blueberries, (4) water), and therefore, randomization of the interventions was applied using Excel. Thus, within each study arm, a randomized crossover design was applied. Initially, the study included women using hormonal contraceptives. However, these individuals (*n* = 12) were excluded from the final analysis due to significant associations between hormonal contraception medication and the primary biochemical markers. Additionally, data from three participants were not included in the final analysis due to withdrawal. A detailed visual description can be seen in Figure 1 (CONSORT flow diagram) and a detailed CONSORT 2025 checklist is provided in the Appendix A.

Ultimately, in total, 30 participants (study arm one: *n* = 16; study arm two: *n* = 14) completed the study protocol and were included in the final analysis.

Eligibility criteria required participants to be female, with a body mass index (BMI) between 18.5 and 25.0 kg/m^2^, a minimum body fat percentage of 20% and general good health, with no evidence of acute or chronic illness. Participants had to be non-smokers, not engage in more than two hours of physical activity per week and refrain from regular resistance training.

### 2.3. Intervention Procedures and Sample Collection

Following successful completion of medical screening and confirmation of eligibility, participants were enrolled in the study. Utilizing an RCT design with two parallel interventions (study arm one: 1× blueberries, 1× whole-grain bread, 1× bread roll, 1× water (control); study arm two: 1× pomegranate juice, 1× water), each participant underwent multiple interventions, separated by a washout period of at least seven days. The interventions within each study arm were administered in a randomized order. Specifically, participants in study arm one consumed blueberries, bread roll, whole-grain bread and water, while participants in study arm 2 consumed pomegranate juice and water once each. The intervention order was randomized using Excel to minimize potential time effects, which can occur if applications are applied consecutively.

The decision to form two study arms was based on the high time burden for the participants, and we expected a reduced possibility of drop-out.

All intervention sessions were conducted at the Nutrition and Training Laboratory (NuTraLab) at the Department of Nutritional Sciences, University of Vienna, Austria. Each intervention day started at 8:30 a.m. to maintain stable physiological conditions across testing days. At each designated time point, venous blood samples were collected via a butterfly needle inserted into an antecubital vein. Additionally, capillary blood was obtained from the fingertip.

Venous blood was collected into EDTA-coated vacutainers (Greiner Bio-One GmbH, Kremsmünster, Austria) for plasma analyses. Blood samples were centrifuged at 3500 rpm for 15 min at 4 °C to separate plasma and serum fractions. Plasma aliquots were subsequently stored at −80 °C until further biochemical analysis, while capillary blood samples were processed immediately for ROS determination. Room temperature was consistently maintained between 20 and 22 °C during all intervention days to standardize environmental conditions.

### 2.4. Dietary Intervention and Blinding

Following baseline blood collection, participants were instructed to consume the randomly assigned intervention within 10 min and rest for two hours at the study center to allow digestion. The order of interventions was randomized for each participant in a crossover design, using an Excel (Microsoft, Redmond, WA, USA) randomization method. Only the study leader had access to the random allocation sequence. Data analysts were blinded during analysis. Study participants received a pseudonymous code, and all blood samples were marked with this code.

The dietary interventions consisted of 250 mL of pomegranate juice, 125 g of blueberries, a 60 g white bread roll, 90 g of whole-grain bread and 250 mL of tap water, serving as a control. These food serving sizes were selected as they reflect realistic portion sizes of adults.

Blueberries, bread rolls and whole-grain bread were purchased daily from a local supermarket and bakery. The pomegranate juice was purchased in a single batch and stored at 4 °C to minimize batch variability. All selected food items and beverages were of organic quality to reduce the presence of additives and simplify ingredient composition.

Each intervention was consumed after a 12 h overnight fasting period and two hours prior to performance of a standardized resistance-circuit HIIT session. Participants were instructed to maintain their habitual dietary patterns and physical activity levels (≤2 h per week) for the duration of the study until all assigned intervention days were completed. The use of dietary supplements was prohibited throughout the study period.

A detailed description of the nutritional value of the applied intervention foods and beverages is presented in Table 1.

### 2.5. Resistance-Circuit Training Protocol (HIIT)

To ensure an adequate and consistent training stimulus across all intervention days, the resistance load was individually adjusted based on performance. If a participant was able to complete at least 60 repetitions for a given exercise during a session, the load for that exercise was increased by 5% on the subsequent intervention day. This adjustment was made to maintain a high level of exertion on each single study day according to Guidelines of the American Journal of Sports Medicine (ACSM), which recommends increasing training load by 2–10% if an individual exceeds the target repetition range [27].

Due to the crossover study design, including a minimum 7-day washout period between sessions and the restriction of additional resistance training during the study period, the likelihood of systematic strength improvements was minimized.

Prior to each HIIT session, participants completed a standardized warm-up, consisting of joint mobilization exercises and a preparatory circuit performed at 25% of the individual one-repetition maximum (1-RM) to stimulate blood flow and prepare the musculoskeletal system—particularly the joints and tendons—for the upcoming HIIT.

The HIIT protocol included three sets of the following exercises (50% of 1-RM, 40 s per set): rowing, chest press, leg curl, lat pulldown and leg press.

To evaluate the intensity of the HIIT, the BORG scale was used as a subjective evaluation [28].

### 2.6. Laboratory Measurements

#### 2.6.1. Reactive Oxygen Species (ROS)

ROS production was assessed via electron paramagnetic resonance (EPR) spectroscopy (Bruker, Ettilingen, Germany). At each time point, 12.5 µL of capillary blood was collected from the fingertip and mixed with 12.5 µL of oxygen-sensitive label (Noxygen, NOXX15.1, 5 µmol/L) and 25 µL of CMH spin probe diluted in Krebs-HEPES buffer (Noxygen, 400 µmol/L). After vortexing, 40 µL of the mixture was transferred into a capillary tube (Hirschmann, Eberstadt, Germany) and incubated for 60 s at 37 °C using a temperature- and gas-controlled EPR system (Noxygen, Elzach, Germany).

The CMH probe reacts with ROS both intra- and extracellularly to form a stable radical CM • [29], which was detected using the following EPR settings: center field 3489.510 G, sweep width 60.0 G, frequency 9.779 GHz, power 20.97 mW, modulation amplitude 2.13 G, sweep time 5.24 s, and modulation frequency 86 kHz. ROS levels were expressed as µmol/min.

#### 2.6.2. Ferric Reducing Ability of Plasma (FRAP)

To assess total antioxidative capacity of plasma, the ferric reducing ability of plasma assay (FRAP) was applied based on the method of Benzie, 1996 [30]. For quantification of antioxidative capacity of plasma samples, an FeSO_4_·7H_2_O standard series (range 100 µM to 2000 µM) was arranged. TROLOX, a vitamin E analog, was applied as the positive control. Before analysis, plasma samples were thawed from −80 °C. For performance of the assay, 10 µL of EDTA plasma samples or 10 µL of FeSO_4_·7H_2_O standards, 30 µL of distilled water and 300 µL FRAP reagent (25 mL 300 mM acetate buffer, 2.5 mL 10 mM TPTZ solution, 2.5 mL 20 mM FeCl_3_·6H_2_O) were applied to the microplate. Incubation for six minutes at 37 °C was applied. Subsequently, absorbance of the EDTA samples was determined at 595 nm (CLARIOstar-plus microplate reader, BMG LABTECH, Ortenberg, Germany). FRAP results were expressed as mM TEAC.

#### 2.6.3. Polyphenol Detection of Intervention Foods and Beverages

Total polyphenol content determination in pomegranate juice

The total polyphenol content (TPC) assessment was adapted from the Folin–Ciocalteu (FC) method [31]. Dilution sets (1:10 for PGJ) were prepared for each sample with distilled water, and gallic acid was used as a standard (1.95 to 1000 µg/mL). Then, 250 µL of the standards and 10 µL of the juice dilutions, as well as 240 µL of distilled water, were transferred to a 96-well microplate. Afterwards, 5 µL of FC reagent and 10 µL of sodium carbonate (Na_2_CO_3_, 350 g/L) were added to every well, followed by 60 min of incubation in the dark at room temperature. Absorbance was measured at 752 nm. The absorbance of each standard was plotted against its concentration. All assays were measured in triplicate. Results were expressed as microgram per gallic acid equivalent per milliliter (µg GAE/mL).

Blueberry Sample Preparation

Fresh blueberries were ground, pre-frozen at −80 °C for two hours and freeze-dried for two days. Sample preparation for the polyphenol content of blueberries was performed according to [32]. First, 25 mg of freeze-dried blueberries was extracted with 5 mL of acetone. The solution was shaken for 30 min at 40 °C using a rotary shaker, centrifuged for 5 min at 1600 rcf and then decanted. This procedure was repeated two more times. The solvent was removed from the combined extracts by evaporation under a gentle stream of N2. The samples were reconstituted in 10 mL of MeOH (80%, *v*/*v*), filtered through a 0.45 μm PVDF syringe filter and stored at 4 °C until further analysis.

Bread roll and whole-grain sample preparation

Both wheat bread rolls and whole-grain spelt rolls were treated the same during sample preparation. First, both were cut in half and dried in the oven for 24 h at 40 °C. After pulverization in a blender, 1 g of ground sample was extracted with 10 mL 80% MeOH (*v*/*v*) at 37 °C for 2 h in a shaker and then centrifuged at 2000 rcf for 5 min. In total, 4 replicates were tested for each bread. The extracts were filtered through a 0.45 μm PVDF syringe filter and stored at 4 °C until further analysis [33]. The wheat roll extracts were diluted 1:2, and the whole-grain spelt roll extracts were diluted 1:4 prior to analysis. Polyphenol content was determined using Folin–Ciocâlteu reagent (Sigma Aldrich, Steinheim am Albruch, Germany).

Total polyphenol content was determined as described by Herchi [34]. In brief, to 2.5 mL of sample extract, 0.5 mL Folin–Ciocâlteu reagent (2 N) was added. After 3 min, 1 mL of 35% Na_2_CO_3_ solution was added, and H_2_O was added to bring the volume to 10 mL before incubating the samples for 1 h at RT in the dark. The solution was then centrifuged for 5 min at 1200 rcf. Total polyphenol content was determined by measuring the extinction at 765 nm. Quantification was performed using a gallic acid standard calibration. Gallic acid standards and blanks were treated the same as the food sample extracts. Here, a 1 mg/mL stock solution of gallic acid was prepared. For that, 11.1 mg/mL gallic acid monohydrate was dissolved in 10 mL of MeOH (80%). The results are expressed as mg gallic acid equivalent per gram dry weight (dw) of the foods (mg/100 g GAE).

### 2.7. Statistical Analysis

A power analysis was conducted (https://www.stat.ubc.ca/~rollin/stats/ssize/n2.html, accessed on 1 September 2021) based on the effect size estimates from a pilot study with a comparable design [25]. Assuming an alpha level of 0.05 and a desired statistical power of 0.80, the calculation indicated that a minimum of 13 participants per group was required to detect significant effects.

To account for the two independent experimental series (blueberries, whole-grain bread, bread roll, water (study arm one) and pomegranate juice, water (study arm two), water baseline, water pre-HIIT to post-HIIT (%-change) and water post-HIIT to 15-post-HIIT (%-change) values from study arm one and study arm two were compared using independent t-tests. As no significant differences were observed (*p* > 0.05), water trials were pooled as a common reference. Functionally similar conditions, meaning no statistically significant differences (*p* > 0.05) between the high-carbohydrate group and the high-polyphenol group, were averaged per participant, yielding two combined test categories.

Statistical analysis was performed using repeated-measures analysis of variance (ANOVA) to assess differences across time points and/or treatment conditions within the same subjects. When appropriate, Mauchly’s test was conducted to assess the assumption of sphericity; if violated, degrees of freedom were corrected using Greenhouse–Geisser adjustment. To further evaluate the effect of the intervention, comprising the carbohydrate group (bread roll, whole grains), polyphenol group (pomegranate juice, blueberries) and water (combined data of water in study arm one and study arm two), a one-way ANOVA was performed on %-changes from pre-HIIT to post-HIIT (to assess acute response to HIIT) and from post-HIIT to 15-post-HIIT (to assess recovery). Post hoc comparisons were conducted using Bonferroni-corrected pairwise tests. Data are presented as mean ± standard error of the mean (SEM), and statistical significance was set at *p* < 0.05. All analyses were conducted using SPSS 28 (IBM).

Furthermore, the 24 h food recalls were analyzed with Nut.s software (dato Denkwerkzeuge, Vienna, Austria; https://www.nutritional-software.at/content/ueber-uns/, accessed on 1 February 2024), and potential associations between vitamin E (mg), vitamin C (mg), beta-carotene (mg), carbohydrate (g) and energy intake (kcal) and ROS (µmol/min) and FRAP (mM TEAC) concentrations at baseline were statistically analyzed using Spearman correlation.

Confounding analysis with fasted baseline levels of 17-beta estradiol was performed using Spearman correlation on baseline, %-change from pre-HIIT to post-HIIT and %-change from post-HIIT to 15-post-HIIT levels of ROS and FRAP.

## 3. Results

After exclusion of participants using hormonal contraceptives, no indications of hormonal confounding via 17-beta estradiol (pg/mL, serum, chemiluminescent microparticle immunoassay), neither on baseline values of FRAP or ROS nor on the %-change in these parameters from pre-HIIT to post-HIIT and post-HIIT to 15-post-HIIT (*p* > 0.05) using Spearman correlation, were observed in the analyzed sample.

No significant associations between vitamin E, beta-carotene, vitamin C, carbohydrate intake and total energy intake (kcal) and baseline FRAP and ROS concentrations were observed by applying Spearman correlation (*p* > 0.05).

No significant differences via paired-t-tests were observed between water from study arm one and study arm two at baseline (ROS: water study arm one vs. study arm two +0.018 µmol/min (*p* = 0.850); FRAP: water study arm one vs. water study arm two +7.5 mM TEAC (*p* = 0.801), %-change pre-HIIT vs. post-HIIT (ROS: water study arm one vs. water study arm two: −4.919% (*p* = 0.212); FRAP: water study arm one vs. water study arm two: −0.315% (*p* = 0.714)) and %-change from post-HIIT to 15-post-HIIT (ROS: water study arm one vs. water study arm two: −4.926% (*p* = 0.194); FRAP: water study arm one vs. water study arm two: −2.351% (*p* = 0.133)).

Absolute concentrations of both ROS (µmol/min) and FRAP (µM TEAC) from baseline to 15 min post-HIIT are presented in Table 2, showing an overall increase in both markers from pre-HIIT to post-HIIT.

### 3.1. Results: Changes from Pre-HIIT to Post-HIIT

As presented in Table 3, in the water control group, ROS (+12.29%) and FRAP (+3.93%) increased from pre-HIIT to post-HIIT, indicating that the exercise protocol posed a considerable physiological challenge.

In this context, the elevation in FRAP primarily reflects a stress-related response, as this assay includes non-enzymatic antioxidant components such as uric acid, albumin and bilirubin, which are known to rise acutely following strenuous exercise [30].

Consumption of carbohydrate-containing foods prior to exercise attenuated the acute HIIT-induced changes in ROS (vs. polyphenol group: −7.75% (*p* = 0.095); vs. water: −2.17% (*p* > 0.05), presented in Figure 2) and FRAP (vs. polyphenol group: −2.16% (*p* < 0.05); vs. water: −2.49% (*p* < 0.01)), suggesting a protective effect against oxidative stress (Figure 3).

### 3.2. Results: Recovery Phase (Δ Post-HIIT **→** 15 min Post-HIIT (%-Change))

%-Changes in ROS and FRAP from post-HIIT to 15 min post-HIIT are presented in Table 4.

The intake of polyphenol-rich foods significantly enhanced recovery following HIIT. Compared to the carbohydrate group, a significant reduction in ROS levels was observed in the polyphenol group (vs. carbohydrates: −8.22% (*p* < 0.01); vs. water: −3.56% (*p* > 0.05)) during the immediate post-exercise period (Figure 4).

No significant group differences were found for changes in FRAP during the recovery period (Figure 5). The continuous increase in FRAP post-exercise likely reflects the high intensity of the exercise session, consistent with previous findings [35].

### 3.3. BORG Scale Results

As presented in Table 5, the BORG scale of perceived exertion showed similar levels in perception of HIIT intensity, indicating a continuous level of intensity throughout the several intervention days.

## 4. Discussion

To the best of our knowledge, this is the first randomized controlled crossover trial (RCT) to compare the acute effects of polyphenol-rich foods (blueberries, pomegranate juice) and carbohydrate-rich foods (whole-grain bread, bread roll) on resistance-circuit high-intensity interval training (RC-HIIT)-induced oxidative stress in healthy sedentary young women.

Preclinical research, largely based on in vitro and animal models, has provided valuable insights into the antioxidant and anti-inflammatory potential of dietary polyphenols [36,37]. However, differences in physiology between humans and animals, combined with the inability of cohort studies to establish causality, underscore the need for controlled human intervention studies. Our trial contributes to closing this gap.

Previous human studies have demonstrated the antioxidative ability of pomegranate juice, where prolonged supplementation (200–500 mL daily for several weeks) significantly reduced indirect markers of oxidative stress [38,39,40]. While informative, these protocols often required high and sustained intakes that are not representative of typical dietary behavior. In contrast, the present study investigated the acute, single-dose effects of polyphenol-rich foods under a controlled exercise-induced stress model.

Evidence from both in vitro and in vivo studies suggests that blueberries exert anti-inflammatory and antioxidative effects by modulating inflammatory markers and reducing ROS [41,42,43,44]. McAnulty demonstrated improved oxidative profiles with six weeks of daily blueberry intake and pre-exercise supplementation before prolonged running [45]. Extending this evidence base, our findings show that polyphenol-rich foods accelerate post-exercise recovery of ROS levels in a resistance-based HIIT context, a training modality less frequently investigated than endurance exercise [46].

The observed faster normalization of ROS following polyphenol intake may be beneficial, as sustained oxidative stress is associated with impaired recovery, reduced muscle function and increased risk of tissue damage [47,48,49]. While a transient increase in ROS is necessary for redox-sensitive signaling and adaptive responses such as hypertrophy and mitochondrial biogenesis [50,51], excessive accumulation compromises contractile function [52]. Thus, nutritional strategies that promote physiological ROS dynamics while preventing overaccumulation may optimize both recovery and adaptation.

Interestingly, carbohydrate consumption in our study attenuated the acute rise in ROS and FRAP. This aligns with prior evidence that sufficient carbohydrate availability reduces exercise-induced oxidative stress by lowering reliance on fat oxidation and minimizing mitochondrial ROS production [53]. Such effects are particularly relevant for endurance and team sport athletes, for whom carbohydrate intake not only sustains performance but also mitigates cumulative oxidative stress during repeated sessions. However, over-suppression of ROS may blunt training adaptations [24], highlighting the importance of dosage in carbohydrate strategies.

### 4.1. Practical Recommendations for Athletes

The presented findings provide practical insights for both athletes and coaches to optimize both performance and long-term training adaptations through nutrition. The observed effects of polyphenol-rich foods and carbohydrate-rich foods on oxidative stress dynamics suggest that these nutrients should be strategically timed according to the athlete’s goals, either competition readiness or training adaptation.

For competitive scenarios or dense training periods, where rapid recovery and sustained performance are prioritized, acute intake of polyphenol-rich foods such as blueberries and pomegranate juice may be advantageous. By accelerating post-exercise recovery of ROS, polyphenols facilitate a quicker return to physiological balance, potentially reducing fatigue, muscle soreness and risk of oxidative damage [54,55].

Similarly, carbohydrate intake before or during high-intensity sessions can attenuate excessive ROS formation while ensuring optimal energy availability [56], thereby preserving performance during repeated bouts or tournaments. In these contexts, suppressing oxidative stress is desirable to maintain training quality and minimize accumulated stress.

In contrast, during general training phases aimed at improving fitness, hypertrophy or metabolic adaptations, excessive suppression of ROS should be avoided. Transient elevations in ROS act as essential signaling molecules driving mitochondrial biogenesis and muscle remodeling [57]. Regular use of high-dose polyphenol supplementation or frequent carbohydrate-driven ROS suppression may blunt these adaptive pathways [24] if consumed habitually around every training session. Instead, athletes might benefit from prioritizing carbohydrate intake primarily for high-volume endurance training, while reserving polyphenol supplementation for recovery weeks or competitions.

From a practical standpoint, whole-food sources of polyphenols, such as berries, pomegranate juice, cherries or dark-colored fruits, can be integrated into pre- or post-exercise meals. Carbohydrate strategies should remain individualized, based on training intensity, duration and energy demands. Together, these findings emphasize the value of tailoring nutrition not only to the type of exercise performed but also to the balance between immediate recovery needs and long-term training objectives.

### 4.2. Limitations

This study has several limitations. First, participants were exclusively sedentary women. Therefore, the generalizability of the findings to physically active individuals or trained athletes is limited, as regular exercise increases endogenous antioxidant capacity and may dampen detectable nutritional effects [58].

Second, all participants were young, healthy females, a group often underrepresented in exercise research; extending the findings to males, therefore, requires caution.

The grouping of whole-grain bread and bread rolls into a single high-carbohydrate, low-polyphenol category, and the grouping of blueberries and pomegranate juice as polyphenol-rich foods, may be subject to criticism. However, our analyses showed that polyphenol-rich interventions most effectively supported post-exercise ROS recovery, independent of their carbohydrate content. Although polyphenols were superior to carbohydrate-only foods, reductions in ROS compared with water were modest and not statistically significant, which reflects the variability typical of fasted HIIT protocols. Differences in fiber and micronutrient content between whole-grain bread and bread rolls did not translate into meaningful physiological differences, supporting their combined classification. Notably, polyphenol-rich foods did not prevent the initial ROS rise from pre- to post-HIIT; the highest acute increases occurred after pomegranate juice and blueberries, indicating that their primary effects relate to recovery rather than to immediate ROS mitigation.

This pattern is noteworthy given the comparable carbohydrate contents of the pomegranate juice and bread conditions. One possible explanation is that polyphenol-rich foods provide a substantially higher reducing capacity, which may transiently participate in redox cycling under the pronounced oxidative flux of fasted HIIT. Such compounds could increase ROS turnover during exercise rather than blunt ROS formation, whereas carbohydrate-rich foods mainly exert energetic effects with slower absorption and lower acute redox reactivity. During recovery, however, the polyphenol-derived reducing milieu may facilitate a faster normalization of ROS levels, consistent with the more pronounced post-exercise decline observed in our data.

Additionally, we recommend including a more detailed assessment of dietary intake prior to the intervention days in trials that examine polyphenol consumption. The nutritional software available to us did not allow for the quantification of dietary polyphenols. Therefore, future studies should consider using software capable of accurately analyzing specific phytochemical subclasses, including polyphenols. Alternatively, participants could be provided with a standardized low-polyphenol diet for 24 h before each trial day to ensure a consistent food matrix and further reduce the risk of potential confounding. Furthermore, we would also recommend extending the nutritional confounding analysis to protein and fat intake as well, as we only focused on antioxidative vitamins, carbohydrates and overall caloric intake before each trial day.

Furthermore, the treatment allocation shown in Figure 1 was not ideally balanced, with four interventions in study arm one and two interventions in study arm two. This resulted from the high participant burden associated with the study procedures, which included intensive exercise sessions, repeated blood sampling and substantial time commitments. These constraints limited how many interventions each participant could safely complete, and the chosen allocation helped minimize fatigue and increase the possibility that a high number of participants would successfully finish all intervention days. Nevertheless, we acknowledge that a more balanced design could further improve methodological rigor, and we therefore recommend that future trials consider more even group distributions, such as three vs. three, or ideally, a single cohort completing all interventions if participant burden allows.

### 4.3. Rationale for Selecting the Nutritional Interventions

Furthermore, we would like to further emphasize the rationale behind selecting pomegranate juice, blueberries, whole-grain bread and white bread rolls as dietary interventions in the present study.

Pomegranate juice has been reported to exhibit strong antioxidative activity in vitro [59] and to reduce indirect markers of oxidative stress such as malondialdehyde (MDA) in a human RCT [60]. However, to our knowledge, its direct and acute effects on reactive oxygen species (ROS) formation in vivo have not yet been investigated.

Similarly, in vitro studies with blueberries have demonstrated promising antioxidative effects [61]. Nevertheless, human intervention studies on their antioxidant potential have produced heterogeneous results, as summarized in a recent systematic review [62]. We therefore considered it important to contribute additional evidence regarding the acute effects of blueberry consumption on oxidative stress and their potential relevance to human health.

In contrast, previous human trials comparing whole grains and refined grains have yielded inconsistent findings. A 14-day intervention by Enright [63] reported no significant differences in oxidative stress biomarkers between whole grains and refined grains. To further explore this, we aimed to determine whether an acute, single consumption of whole-grain bread or refined-grain bread would elicit measurable differences in oxidative stress markers. As no such differences were observed, these foods were collectively categorized as high-carbohydrate, low-polyphenol foods. Nevertheless, whole grains remain a notable source of phytochemicals with antioxidative potential [64], and the acute physiological distinctions between whole-grain bread and refined grains warrant further investigation.

## 5. Conclusions

This study demonstrates that polyphenols and carbohydrates play distinct yet complementary roles in modulating exercise-induced oxidative stress and recovery. Acute intake of polyphenol-rich foods, such as blueberries and pomegranate juice, supported the restoration of redox balance following resistance-circuit HIIT, indicating benefits especially for short-term recovery in settings with elevated training frequency or competition demands.

Conversely, carbohydrate ingestion attenuated the acute rise in oxidative stress during exercise, likely reflecting energetic support and a shift toward carbohydrate oxidation. While this may help sustain performance in repeated or prolonged efforts, the implications for long-term training adaptations warrant further investigation.

Overall, these findings highlight dietary polyphenols as a feasible whole-food strategy to support post-exercise recovery, whereas carbohydrate-rich foods may help modulate acute oxidative responses during exercise. Future studies should determine how timing, dose and combinations of these nutrient groups can best balance immediate recovery with the preservation of redox-sensitive training adaptations.

Furthermore, investigating whether combinations of the tested foods yield additive or synergistic effects in real-world dietary patterns may improve our understanding of how whole-diet composition modulates redox homeostasis. Finally, clarifying how antioxidant-rich foods interact with components such as saturated fats, refined sugars or highly processed foods—dietary factors with potential detrimental impact on oxidative parameters [65]—may reveal whether dietary enrichment with potential beneficial foods or dietary restriction of potential detrimental foods influences oxidative balance and long-term health.

## Figures and Tables

**Figure 1 antioxidants-14-01481-f001:**
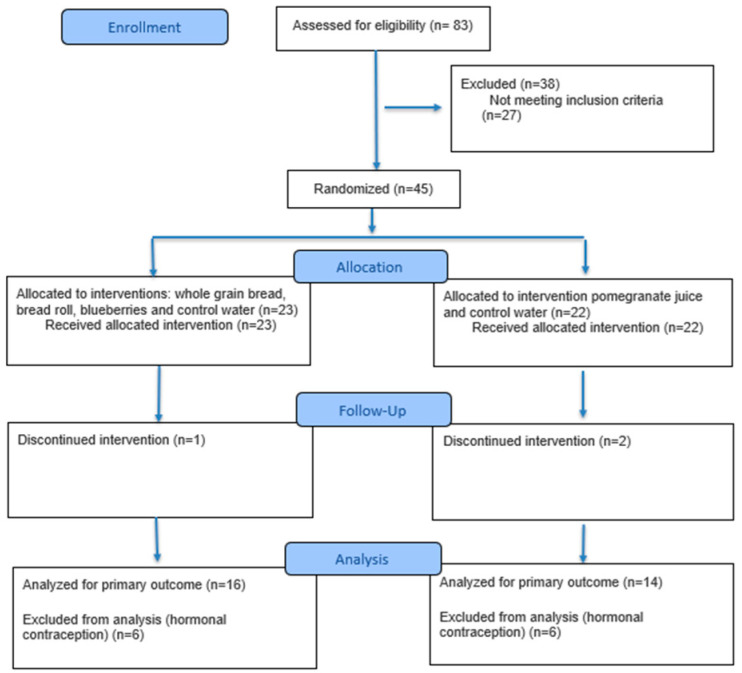
CONSORT flow diagram. https://www.consort-spirit.org/.

**Figure 2 antioxidants-14-01481-f002:**
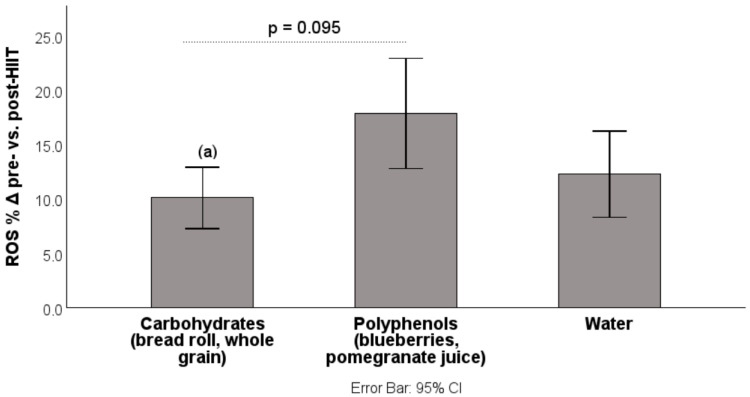
Percentage increase in ROS from pre-HIIT to post-HIIT. ROS: reactive oxygen species; HIIT: high-intensity interval training; data are presented as mean ± CI. (a) Different from polyphenol group (−7.75%, *p* = 0.095, Bonferroni post hoc test, trend).

**Figure 3 antioxidants-14-01481-f003:**
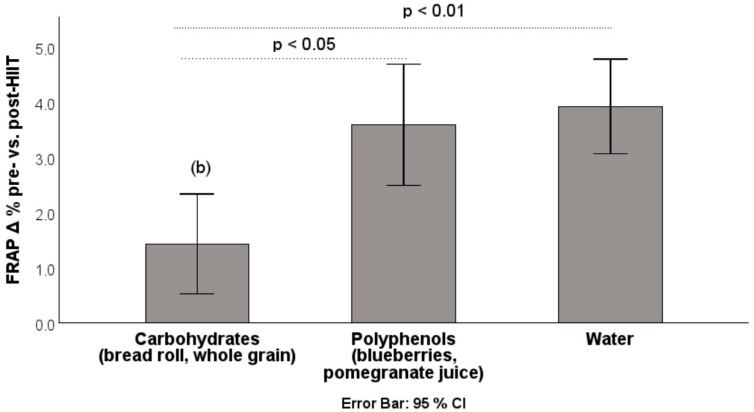
Percentage increase in FRAP from pre-HIIT to post-HIIT. FRAP: ferric reducing ability of plasma; HIIT: high-intensity interval training; data are presented as mean ± CI. (b) Significantly different from polyphenols (−2.16%, *p* < 0.05) and water (−2.49%; *p* < 0.01) (Bonferroni post hoc test).

**Figure 4 antioxidants-14-01481-f004:**
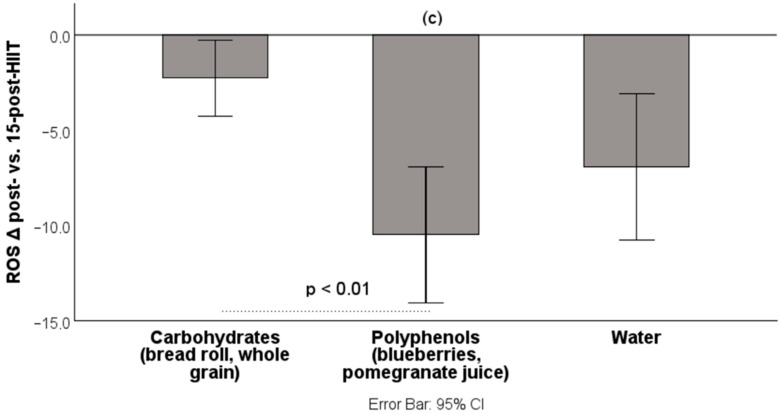
Percentage decrease in ROS from post-HIIT to 15-post-HIIT. ROS: reactive oxygen species; HIIT: high-intensity interval training; data are presented as mean ± CI. (c) Significantly different from the carbohydrate group (−8.22%, *p* < 0.01, Bonferroni post hoc test).

**Figure 5 antioxidants-14-01481-f005:**
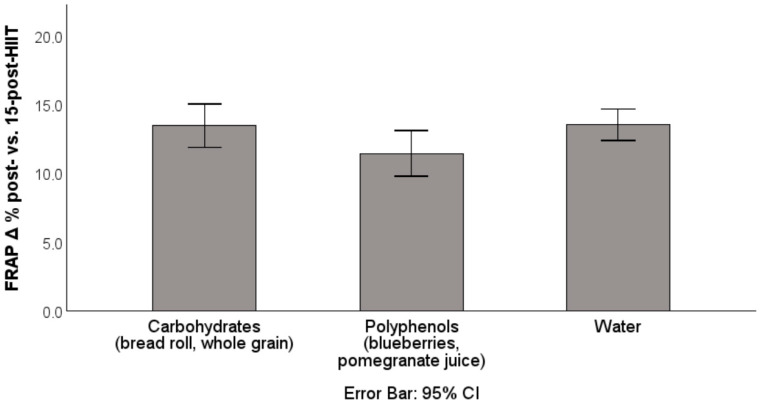
Percentage increase in FRAP from post-HIIT to 15-post-HIIT. FRAP: ferric reducing ability of plasma; HIIT: high-intensity interval training; data are presented as mean ± CI.

**Table 1 antioxidants-14-01481-t001:** Nutritional value of applied intervention foods and beverages.

Nutrients/Portion	Pomegranate Juice	Blueberries	Whole-Grain Bread	Bread Roll
Portion-size	250 mL	125 g	90 g	60 g
Energy (kcal)	160.0	62.5	245.7	207.4
Fat (g)	<1.3	0.9	4.50	2.2
- of which SFAs (g)	<0.3	0.4	0.1	0.3
Carbohydrates	35.0	10.6	38.5	40.0
- of which sugars (g)	35.0	9.5	1.8	0.3
Fiber (g)	2.8	5.0	6.48	2.3
Protein (g)	<1.3	0.88	9.63	5.7
Sodium (g)	<0.1	<0.1	1.5	1.1
Polyphenol content (mg)	965.5	552.5	174.6	29.00

SFAs: saturated fatty acids.

**Table 2 antioxidants-14-01481-t002:** Absolute concentrations of ROS and FRAP from baseline to 15 min post-HIIT.

Marker	Intervention	Baseline	Pre-HIIT	Post-HIIT	15 min Post-HIIT	T	T × G	G
ROS (µmol/min)	Blueberries	1.00 ± 0.39	0.84 ± 0.25	0.97 ± 0.33	0.93 ± 0.27	<0.001	0.123	0.536
	Bread roll	1.00 ± 0.26	0.82 ± 0.21	0.93 ± 0.27	0.91 ± 0.26			
	Pomegranate	1.09 ± 0.22	0.93 ± 0.18	1.18 ± 0.27	0.99 ± 0.28			
	Whole Grain	0.95 ± 0.27	0.87 ± 0.24	0.93 ± 0.26	0.91 ± 0.22			
	Water	0.98 ± 0.25	0.91 ± 0.21	1.03 ± 0.29	0.94 ± 0.22			
FRAP [mM TEAC]	Blueberries	828.4 ± 68.8	812.0 ± 80.0	841.0 ± 77.0	932.8 ± 75.3	<0.001	0.346	0.510
	Bread roll	803.2 ± 80.1	771.2 ± 76.2	788.6 ± 77.3	892.5 ± 69.9			
	Pomegranate	830.7 ± 72.5	811.6 ± 71.2	843.8 ± 82.7	926.8 ± 86.5			
	Whole Grain	822.3 ± 73.4	807.5 ± 83.1	811.7 ± 75.0	924.2 ± 80.5			
	Water	806.6 ± 79.3	783.1 ± 67.2	818.6 ±78.8	928.3 ± 92.1			

T: Time; T × G: Time × Group; G: Group; FRAP: ferric reducing ability of plasma; ROS: reactive oxygen species.

**Table 3 antioxidants-14-01481-t003:** Changes in ROS and FRAP from pre-HIIT to post-HIIT (%).

Intervention	Δ ROS Pre → Post %	Δ FRAP Pre → Post %
Blueberries (*n* = 16)	+12.96 ± 10.52	+2.59 ± 2.41
Pomegranate J. (*n* = 14)	+23.46 ± 14.71	+4.75 ± 3.16
Bread roll (*n* = 16)	+12.53 ± 9.13	+1.68 ± 3.13
Whole grains (*n* = 16)	+7.69 ± 5.48	+1.20 ± 1.75
Water (*n* = 30)	+12.29 ± 10.63	+3.93 ± 2.29

ROS: reactive oxygen Species; FRAP: ferric reducing ability of plasma; HIIT: high-intensity interval training; data are presented as mean ± SD.

**Table 4 antioxidants-14-01481-t004:** ROS and FRAP changes from post-HIIT to 15 min post-HIIT.

Intervention	Δ ROS Post vs. 15 Post %	Δ FRAP Post—15 Post %
Blueberries	−7.41 ± 5.72	+11.44 ± 4.83
Pomegranate juice	−13.99 ± 11.83	+11.37 ± 4.08
Bread roll	−2.52 ± 4.42	+13.33 ± 3.89
Whole grains	−2.00 ± 6.57	+13.49 ± 5.00
Water	−6.92 ± 10.24	+13.47 ± 3.05

HIIT: High-intensity interval training; ROS: reactive oxygen species; FRAP: ferric reducing ability of plasma; data are presented as mean ± SD.

**Table 5 antioxidants-14-01481-t005:** BORG scale of perceived exertion.

Intervention	BORG Pre-HIIT	BORG Post-HIIT	Δ % Pre–Post HIIT
Blueberries	6.06 ± 0.25	16.78 ± 1.54	177.15 ± 27.20
Pomegranate Juice	6.21 ± 0.58	17.68 ± 1.25	186.39 ± 30.46
Bread Roll	6.03 ± 0.13	16.13 ± 1.37	167.47 ± 23.56
Whole Grain	6.06 ± 0.25	16.31 ± 1.66	169.35 ± 28.42
Water	6.22 ± 0.55	17.33 ± 1.57	180.93 ± 35.21

Data are presented as mean ± SD; HIIT: high-intensity interval training.

## Data Availability

The original contributions presented in this study are included in the article and Appendix A. Further inquiries can be directed to the corresponding authors.

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
