# Peer review of "Acute Impact of Polyphenol-Rich vs. Carbohydrate-Rich Foods and Beverages on Exercise-Induced ROS and FRAP in Healthy Sedentary Female Adults—A Randomized Controlled Trial"

_antioxidants, 2025, doi:10.3390/antiox14121481_

Round 1
Reviewer 1 Report
My key concern relates to the profile of foods used. Pomegranate juice has a near equivalent amount of carbohydrate to the “carbohydrate foods” – given this, it is difficult to disentangle the comparison between carbohydrate vs polyphenols. Furthermore, the lack of significance compared to water for many time points is overlooked.
Thanks for the opportunity to review this paper.
I have some comments below, but my key concern relates to the profile of foods used. Pomegranate juice has a near equivalent amount of carbohydrate to the “carbohydrate foods” – given this, it is difficult to disentangle the comparison between carbohydrate vs polyphenols vs water.
Abstract
I found the abstract a little misleading in terms of the design as it isn’t clear that the foods were consumed separately and then data combined for the pomegranate/blueberries and roll/bread. This could be clarified. Also, water control is not stated – alluded to in the results.
Please include timing of consumption relative to exercise and timing of sampling.
Lines 33 - 37 – inconsistent in reporting of comparison to control (water). Should include all comparisons - Noting many were not significant.
Given the NS comparisons against water, I don’t think the conclusion reflects the findings.
Introduction
I’d like to see some mention of the stress response to different forms of exercise – to set up why you chose HIIT as the exercise stimulus.
Method
So, there were 5 possible treatments? This could be clearly stated in 2.1. or in 2.3 – ‘multiple interventions’
Was habitual dietary intake assessed? If not, this is a limitation.
Lines 120-122 – study participants. The order is confusing. Mention enrolling 42 participants but after that say 45 were randomly assigned. Please clarify.
Given the average age of the female participants, was ovulatory status measured in any way? My concern here revolves around how menstrual cycle influences ROS.
I gleamed from the results that data from foods were combined -pomegranate/blueberries and roll/bread – this needs to be mentioned in the stats. Similarly, it is unclear why two breads were evaluated - noting one has more fibre – but their data were combined - seems unnecessarily complicated?
30 participants in the final analysis and later it is mentioned that they didn’t complete every intervention condition. What were the final numbers for each condition? How were these missing data handled – my understanding of ANOVA is that a participant will be dropped if data are incomplete. There are other statistical approaches that account for missing data.
Results
Table 2 is not mentioned in the results – should summarise findings from this table.
I found it frustrating that data were reported individually for each food in tables but then in combination in the figures and in text (especially as there was no mention in method that this was how data would be evaluated). Could just include these combined values in the tables. But I also think this type of analysis is misleading given the carbohydrate content of the pomegranate juice.
Additionally, I am unclear why two breads with the same carbohydrate content were evaluated if they’re combined in the results?
Discussion
The carbohydrate content of the pomegranate juice is a major limitation that needs to be addressed and considered in the interpretation of the results. I.e. Why did pomegranate juice have the highest ROS if carbohydrate is beneficial?
And how does this all relate to the water…which statistically was equally effective in some instances?
Additionally, I am unclear why two breads with the same carbohydrate content were evaluated? The obvious difference is fibre, but the nutrient difference isn’t discussed, so why was this done?
Author Response
Dear Reviewer,
We thank you for your thorough and constructive feedback. We have made every effort to address all of your comments as comprehensively as possible.
………………………….
Comment 1:
“My key concern relates to the profile of foods used. Pomegranate juice has a near equivalent amount of carbohydrate to the “carbohydrate foods” – given this, it is difficult to disentangle the comparison between carbohydrate vs polyphenols. Furthermore, the lack of significance compared to water for many time points is overlooked.”
Response 1:
We aimed to test two carbohydrate sources with different glycemic indices (GI) in order to detect potential differences in absorption and/or glycemic response. Interestingly, no difference was observed. We further assume that the lack of effect of the sugar in the pomegranate juice might be related to differences in absorption between beverages and solid foods.
……………………..
Abstract
Comment 2:
“I found the abstract a little misleading in terms of the design as it isn’t clear that the foods were consumed separately and then data combined for the pomegranate/blueberries and roll/bread. This could be clarified. Also, water control is not stated – alluded to in the results.”
Response 2:
We have now clarified that the foods were consumed separately and have specified the inclusion of the water control condition.
……………………
Abstract
Comment 3:
“Please include timing of consumption relative to exercise and timing of sampling.”
Response 3:
We have included the timing of consumption relative to exercise and timing of sampling.
………………….
Abstract
Comment 4:
“Lines 33 - 37 – inconsistent in reporting of comparison to control (water). Should include all comparisons - Noting many were not significant.”
Response 4:
We have included in the abstract that many of the comparisons did not differ significantly from the water control.
……………….
Abstract
Comment 5:
“Given the NS comparisons against water, I don’t think the conclusion reflects the findings.”
Response 5:
We would like to emphasize why, from our perspective, the conclusion that polyphenols are more effective than carbohydrates in promoting recovery represents a substantial finding:
Polyphenols are generally recognized for their antioxidant and anti-inflammatory potential. However, under certain conditions they may also exhibit pro-oxidant behavior. In the presence of transition metals such as Fe2+ or Cu+, many phenolic compounds can undergo redox cycling, thereby reducing metal ions and promoting the Fenton reaction that generates hydroxyl radicals. This phenomenon is particularly relevant under acute inflammatory conditions, where free metal ions and ROS are abundant, shifting the redox balance toward oxidative stress rather than protection. High concentrations of flavonoids such as quercetin, EGCG or gallic acid have been shown to induce hydrogen peroxide formation and DNA damage in vitro. Such context-dependent duality reflects the “antioxidant paradox”, whereby compounds with antioxidant properties can also act as pro-oxidants depending on concentration, pH, metal availability and cellular redox state. Hence, the biological impact of polyphenols must be interpreted in light of their environment and dose-dependent redox behaviour.
Halliwell, B. (2008). Are polyphenols antioxidants or pro-oxidants? What do we learn from cell culture and in vivo studies?. Archives of biochemistry and biophysics, 476(2), 107-112.
……………
Introduction
Comment 6:
“I’d like to see some mention of the stress response to different forms of exercise – to set up why you chose HIIT as the exercise stimulus.”
Response 6:
We chose resistance circuit HIIT as the exercise stimulus since it is less explored than conventional aerobic protocols and resistance training.
To our knowledge, so far the only two resistance based circuit HIIT protocols to generate a stress reaction that have been used are those from Zeng and Bizheh. We found those protocols interesting and were inspired by them.
Zeng, Z., Jendricke, P., Centner, C., Storck, H., Gollhofer, A., & König, D. (2020). Acute effects of oatmeal on exercise-induced reactive oxygen species production following high-intensity interval training in women: a randomized controlled trial. Antioxidants, 10(1), 3.
Bizheh, N., & Jaafari, M. (2011). The effect of a single bout circuit resistance exercise on homocysteine, hs-CRP and fibrinogen in sedentary middle aged men. Iranian journal of basic medical sciences, 14(6), 568.
…………………………..
Methods
Comment 7
”So, there were 5 possible treatments? This could be clearly stated in 2.1. or in 2.3 – ‘multiple interventions’”
Response 7:
We stated that the study design in Section 2.1 included five treatments.
……
Methods
Comment 8
“Was habitual dietary intake assessed? If not, this is a limitation.”
Response 8:
We have included the description of the 24-hour food recall analysis in the study design section.
……….
Methods
Comment 9
“Lines 120-122 – study participants. The order is confusing. Mention enrolling 42 participants but after that say 45 were randomly assigned. Please clarify.“
Response 9:
We have corrected this and highlighted the change (the number 42 reflects the count after excluding the three dropouts).
……………….
Methods
Comment 10:
“Given the average age of the female participants, was ovulatory status measured in any way? My concern here revolves around how menstrual cycle influences ROS.”
Response 10:
We have now included the description in Section 2.2 (Study participants).
……..
Methods
Comment 11:
“I gleamed from the results that data from foods were combined -pomegranate/blueberries and roll/bread – this needs to be mentioned in the stats. Similarly, it is unclear why two breads were evaluated - noting one has more fibre – but their data were combined - seems unnecessarily complicated?”
Response 11:
We combined the data for certain foods (pomegranate juice/blueberries and bread roll/whole grain bread) to investigate whether differences in fiber content or polyphenol content influenced recovery after exercise. Our findings were notable: bread differing in fiber content (whole grain vs. white bread) appeared to have similar effects, suggesting fiber content may not substantially influence the outcomes observed. Furthermore, polyphenol-rich foods demonstrated robust effectiveness in promoting recovery, independent of their carbohydrate content. These observations provide novel insights into the role of polyphenol intake and carbohydrate type in exercise recovery.
………
Methods
Comment 12:
“30 participants in the final analysis and later it is mentioned that they didn’t complete every intervention condition. What were the final numbers for each condition? How were these missing data handled – my understanding of ANOVA is that a participant will be dropped if data are incomplete. There are other statistical approaches that account for missing data.”
Response 12:
I have included the number of participants for each intervention into table 3,4 and 5.
Only participants who finished both the water control day and at least one food intervention were included into the analysis.
Please find a more detailed description of our statistical procedure in section 2.7 statistical analysis.
……………….
Results
Comment 13:
“Table 2 is not mentioned in the results – should summarise findings from this table.”
Response 13:
We have now referenced table 2 in the results section.
………………….
Results
Comment 14:
“I found it frustrating that data were reported individually for each food in tables but then in combination in the figures and in text (especially as there was no mention in method that this was how data would be evaluated). Could just include these combined values in the tables.”
Response 14:
We aimed to present both individual intervention data and combined categories to provide a complete picture. While individual data are shown in the tables, we found that the combined categories, particularly for polyphenol-rich foods, are more effectively communicated through graphical illustrations, which allow clearer visualization of the differences than would be achieved by including these combined values in the tables. Furthermore, Reviewer 2 requested us to keep the figures and slightly improve their quality.
………………..
Discussion
Comment 15
“The carbohydrate content of the pomegranate juice is a major limitation that needs to be addressed and considered in the interpretation of the results. I.e. Why did pomegranate juice have the highest ROS if carbohydrate is beneficial? And how does this all relate to the water…which statistically was equally effective in some instances?”
Response 15:
We have included a detailed description of our perspective on your concerns in the limitation section of our manuscript.
…………..
Discussion
Comment 16:
“Additionally, I am unclear why two breads with the same carbohydrate content were evaluated? The obvious difference is fibre, but the nutrient difference isn’t discussed, so why was this done?”
Response 16:
We initially found it interesting that there was no substantial difference between whole grain bread and bread roll in our outcomes. Therefore, we combined them into a single category of high-carbohydrate, low-polyphenol foods. This grouping is further supported by considerations of digestibility and glycemic index, as both types of bread are rapidly digested and produce similar postprandial glucose response, despite the common perception that whole grains are superior to refined grains.
Reviewer 2 Report
The authors presented the first randomized controlled crossover trial (RCT) to compare the acute effects of polyphenol-rich foods (blueberries, pomegranate juice) and carbohydrate-rich foods (whole grain bread, bread roll) on resistance circuit high intensity interval training (RC-HIIT). Presented study received ethical approval from the Ethics Committee of the University of Vienna (Approval ID: 00743 -1.12.2021) and was registered on ClinicalTrials.gov under the identifier NCT05242978. (Registration date: 26.1.2022). As a result of the study, the authors presented practical recommendations for athletes and highlighted limitations that should be considered when interpreting the results.
The research conducted and the results presented are valuable. However, the manuscript still requires revisions, which the authors should consider.
- Text formatting needs to be improved. In its current form, the text is difficult to read. Missing, among others: paragraph indents.
- Figure captions should appear below the figures.
- Table 4 - use the minus symbol instead of the dash.
- Figure 1-3: Axes and graphic captions should be larger.
- Graphics and tables should appear in the text after the first citation.
- It would be worthwhile to highlight the selection of specific foods in the manuscript. It would be worthwhile to consider whether their consumption complemented each other or had a synergistic effect? ​​Could other foods included in the diet have had a positive impact or worsened the results?
Author Response
Dear Reviewer
We thank you for your thorough and constructive feedback. We have made every effort to address all of your comments as comprehensively as possible.
……………………………………………
Point by Point Response Reviewer 2
Comment 1:
“Text formatting needs to be improved. In its current form, the text is difficult to read. Missing, among others: paragraph indents.“
Response 1:
We have included paragraph indents to improve clarity.
…….
Comment 2:
“Figure captions should appear below the figures.”
Response 2:
We have revised the manuscript accordingly. All figure captions now appear below their respective figures.
………………
Comment 3:
“Table 4 - use the minus symbol instead of the dash.”
Response 3:
I hope I understand it correctly what you mean by that. I have changed the “Δ FRAP post -> 15 post %” into “Δ FRAP post – 15post %”
……………….
Comment 4:
“Figure 1-3: Axes and graphic captions should be larger.”
Response 4:
We have increased the size of axes and graphic captions in the mentioned figures as suggested.
…………
Comment 5:
“Graphics and tables should appear in the text after the first citation.”
Response 5:
We have revised the manuscript accordingly. All graphics and tables now appear in the text immediately after their first citation.
…………………..
Comment 6:
“It would be worthwhile to highlight the selection of specific foods in the manuscript. It would be worthwhile to consider whether their consumption complemented each other or had a synergistic effect? ​​Could other foods included in the diet have had a positive impact or worsened the results?”
Response 6:
We have expanded the discussion section in line with your suggestions. Specifically, we added a new subsection entitled “Rationale for selecting the nutritional interventions”, in which we provide a more detailed explanation of our choices. In addition, we have incorporated new considerations for future long-term studies at the end of the conclusion section.
…………….
Round 2
Reviewer 1 Report
In assessing the responses there are several new aims and concepts introduced that are not clear in the manuscript. These relate to differences in glycaemic index, fibre, foods vs beverages, reason for HIIT.
I don’t think it is sufficient to put these in a reply but not integrate all elements into the manuscript. There needs to be greater clarity in the introduction around carbohydrates and GI (and contributing factors – ie fibre). Similarly, what is known about HIIT and ROS? Again, this should be touched on in the introduction. And then there is the idea of food vs. beverages which is how the method seems to be designed.
I don’t understand what the true aim was given the study design and analysis. More detail would assist with this.
Abstract
Randomized controlled trial mentioned a couple of times – consider deleting one mention.
Line 29 (and later in method line 102) - The study design is unclear to me – two parallel intervention cohorts, in which participants completed five single-dose intervention days – what were the parallel interventions?
Line 36-38 “No significant differences vs. water were observed for the acute ROS response, which could be expected under fasted HIIT conditions.”
-is acute post-HITT? May be clearer to state post-HITT? The language of acute is suitable as a summary/conclusion but in describing where the differences (did not) occur, this is unclear.
It is always difficult to report close but ultimately non-significant findings, but we need to be mindful of p-value “spin”. To me, concluding ‘CHO mitigated oxidative stress…” (not significant and not different to water) perpetuates p-value spin and this language needs to be softened.
Introduction
Regarding Comment 6:
“I’d like to see some mention of the stress response to different forms of exercise – to set up why you chose HIIT as the exercise stimulus.”
Thanks for presenting your rationale in your reply, but please incorporate into the introduction.
The introduction also needs to mention glycaemic index and fibre effects given you use this to justify the foods chosen (re Comments 1 and 11)
Material and Methods
Line 102 and 159 – please provide a better description of the parallel intervention as it is also presented as a cross-over trial.
Line 134 states “were randomly assigned to the intervention” – it isn’t entirely clear what they were randomised to? Usually with a cross-over trial there are specific sequences that determine the order of multiple treatments. Please clarify.
But in reviewing figure 1, it shows a very unbalanced treatment allocation (4 vs 2 treatments). Why?
Line 122-124: This presents the results of an analysis, so it seems in the wrong place. How were these associations tested? This approach should be described in the statistics section. Were there differences in polyphenol intake, protein and fat? By baseline FRAP and ROS, presuming this is from blood sampling – is this supposed to intimate that polyphenol intake was similar – unclear to me
Re Comment 10, Line 145 – this introduces a new element – how was “hormonal confounding” evaluated?
Line 148 – food arm and beverage arm?? The methodology is inconsistent with the stated aims to compare CHO vs polyphenols.
Lines 301-304
What does P1 and P2 stand for?
Please justify why no differences at baseline (post fast, no intervention food or drink) led you to combine responses to water? Shouldn’t it be comparing the water trials and then combine them?
What is a ’functionally similar condition’ – this needs to be clearly stated because I am unsure if this is food or beverages given method, but suspect it is meant to be CHO and polyphenols? You also allude in the discussion that you combined these based on no significant differences – this is not clear because you mention combining them before the analysis is described. This section needs to be revised to clearly articulate the process and steps that were taken to support combining the different foods.
Lines 311-315
Please specify that the one-way ANOVA was conducted on the combined data for bread/roll and juice/blueberries
For anyone in the ‘beverage’ arm, this sensitivity analysis seems redundant as they have only one data point? Where are the results of these paired t-tests reported - apologies if I have missed this? How were the findings applied (ie. what was the point?)
Results
Line 340-342. Please include the % and p-value for carbohydrate vs water in the text.
Author Response
Dear Reviewer 1,
Please see the attachment.
Kind regards,
the corresponding authors

Round 3
Reviewer 1 Report
Nil
The authors have responded to my concerns. They may wish to review the method for repetition of newly added information but otherwise it is addressed.